# Phytochemical Profiles and In Vitro Immunomodulatory Activities of Extracts Obtained from *Limonium gmelinii* Using Different Extraction Methods

**DOI:** 10.3390/plants12234019

**Published:** 2023-11-29

**Authors:** Dariya Kassymova, Galiya Zhusupova, Vyacheslav Ogay, Aizhan Zhussupova, Kumar Katragunta, Bharathi Avula, Ikhlas A. Khan

**Affiliations:** 1Department of Chemistry and Technology of Organic Substances, Natural Compounds and Polymers, NPJSC Al-Farabi Kazakh National University, Al-Farabi Ave. 71, Almaty 050040, Kazakhstan; 2Stem Cell Laboratory, National Center for Biotechnology, Korgalzhyn Highway 13/5, Astana 010000, Kazakhstan; 3Department of Molecular Biology and Genetics, NPJSC Al-Farabi Kazakh National University, Al-Farabi Ave. 71, Almaty 050040, Kazakhstan; aizhan.zhusupova@gmail.com; 4National Center for Natural Products Research, School of Pharmacy, The University of Mississippi, University, MS 38677, USA; bavula@olemiss.edu (B.A.); kkatragu@olemiss.edu (K.K.); ikhan@olemiss.edu (I.A.K.)

**Keywords:** *Limonium gmelinii*, immunomodulatory activity, ultrasonic-assisted extraction, conventional extraction, cytokines, ELISA

## Abstract

*Limonium (L.) gmelinii* is a valuable pharmacopoeial Kazakhstani plant. Several studies have reported on the various biological activities of the plant. The purpose of our research was to study and compare the extraction yields, immunomodulatory activities, and chemical compositions of extracts from the above-ground parts of *L. gmelinii* obtained via conventional extraction (CE; Extract 1) and ultrasound-assisted extraction (UAE; Extract 2). The extracts were characterized by a considerable number of polyphenols and flavonoids: 378.1 ± 4.5 and 382.2 ± 3.3 GAE mg/g, and 90.22 ± 2.8 and 94.61 ± 1.9 QE mg/g in Extract 1 and Extract 2, respectively. Extract 2 had a slightly higher extraction yield (33.5 ± 2.4%) than Extract 1 (30.2 ± 1.6%). Liquid Chromatography–Diode-Array Detection–Electrospray Ionization–Quadrupole Time-of-Flight Mass Spectrometry (LC-QToF-MS) revealed the presence of 54 biologically active compounds in both extracts. It was shown that the studied extracts stimulate the secretion of TNF-α and IL-6 by intact mouse peritoneal macrophages and splenic lymphocytes, whilst they have an inhibitory effect on the secretion of these cytokines by activated immune cells. Both extracts demonstrated similar patterns of stimulation and inhibition in a splenocyte proliferation assay. Altogether, the *L. gmelinii* extracts obtained via CE and UAE might be suggested as effective immunomodulatory agents. The application of UAE for this purpose seems to be more efficient with a view of obtaining of a highly potent extract in a much shorter time.

## 1. Introduction

*Limonium* (*L.*) Mill is a diverse genus of perennial grasses and shrubs, composed of almost 600 species [1]. A wide range of breeding options, including mixed reproduction (sexual, vegetative, and apomixis), with the occurrence of hybridization and polyploidy, determine the wide cosmopolitan distribution (arid and saline soils, rocky areas, deserts, and steppes) of its representatives along with a diverse spectrum of biological activities [2]. Plants of the genus are broadly used in decorative [1], medicinal [3], and food purposes [4], for the decontamination of soils from heavy metals [5], etc.

The broad spectrum of the proven therapeutic activity of certain species has been the object of numerous studies on the chemical compositions of their extracts and methods of the optimal extraction of biologically active compounds (BACs). For instance, a phenolic-rich ethanolic extract of *L. delicatulum*, a halophyte growing in the salt marshes of the Mediterranean coast, demonstrated significant antioxidant and antimicrobial activities [6]. Its methanolic extract was found to act as an inhibitor of certain enzymes, including cholinesterase [7] and tyrosinase [8]. Another representative of halophytes common in Mediterranean salt marshes, *L. densiflorum,* exhibited notable antioxidant, anti-inflammatory, antimicrobial, antitumor [9], and antiviral activities [10]. Antitumor and immunomodulatory effects were attributed to the polysaccharides of *L. sinense* (Girard) Kuntze [11]. The inhibitory activity of the aqueous extracts of *L. brasiliense* has been demonstrated against xanthine oxidase upon the treatment of hyperuricemia [12], as well as their antibacterial activity against bacteria with multidrug resistance [13]. Secondary metabolites isolated from *L. tubiflorum* were examined as antiviral components targeting SARS-CoV-2 in in silico studies, and some myricetin derivatives were suggested as potential COVID-19 drug candidates [14].

The most well-studied species of the genus *L.* Mill growing in Kazakhstan are *L. gmelinii* (Willd.) Kuntze, *L. myrianthum* (Schrenk) Kuntze, *L. leptophyllum* (Schrenk) Kuntze, and *L. caspium* (Willd.) Gams. Antibacterial, antifungal, antimalarial, and antileishmanial activities were demonstrated by compounds isolated from *L. caspium* [4]; similar effects were shown by secondary metabolites from *L. myrianthum*, *L. leptophyllum*, and *L. gmelinii* [15]. As such, preclinical tests of an extract from *L. gmelinii* roots showed its antioxidant, hepatoprotective, antibacterial, antimutagenic, antitumor, and antiviral properties [16,17]. It has also been shown that extracts from the roots and rhizomes of *L. gmelinii* protect neurons, astrocytes, and cerebral endothelial cells from oxidative and inflammatory responses in vitro and improve motor functions after ischemic stroke in vivo [18]. Data on the antidiabetic, cytotoxic, and anti-inflammatory properties of lignanamides isolated from the ethyl acetate extract of *L. gmelinii* roots [19].

Hence, *Limonium* species are therapeutically important and can be used for various medicinal purposes, the anti-inflammatory and antioxidant properties of which have been better studied, and these are what possibly make them useful in both acute and chronic inflammation.

Recently, plant-sourced phytochemicals have been in the spotlight because they represent a critical part in immunotherapies to combat the spread of cancer, autoimmune diseases, and infection, and they have promising potential for use as immune-enhancing agents and regulators [20]. It is known that an abnormal immune response against individual cells could have serious consequences, such as chronic inflammation leading to the above-mentioned diseases [21].

Macrophages and lymphocytes are known as key players in the development and resolution of inflammation, as they secrete a variety of inflammatory mediators, including several cytokines, inorganic reactive radicals, and reactive oxygen and nitrogen intermediates [22]. TNF-α and IL-6 are important biomarkers of proinflammatory responses; the dysregulation of their production is associated with a variety of inflammatory disorders [22]. An uncontrolled proliferation of lymphocytes can also lead to undesirable conditions like chronic or autoimmune diseases [23]. In general, immunomodulators modify immune responses, which, in turn, may lead to increased or decreased immune reactions [20].

The principal aim of the current research was to study and compare the immunomodulatory effect of *L. gmelinii* extracts obtained using methods of conventional extraction (CE) and ultrasound-assisted extraction (UAE) by measuring the levels of TNF-α and IL-6 produced by murine macrophages and splenic lymphocytes, as well as the proliferation of splenocytes. To the best of the authors’ knowledge, these properties have not yet been investigated in *L. gmelinii* species.

## 2. Results

### 2.1. Preparation of Extracts and Their Standardization

In our study, the efficiency of the CE and UAE methods of extraction was assessed via the yield of extracts and the content of the main secondary metabolites in them (Table 1). According to the yield, the *L. gmelinii* extracts obtained via 45 min ultrasonication (Extract 1) and 5 h double maceration (Extract 2) amounted to 30.2 ± 1.6% and 33.5 ± 2.4%, respectively (Table 1). The total phenolic contents (TPCs) of the extracts obtained via UAE and CE were 378.1 ± 4.5 mg gallic acid equivalent (GAE)/g and 382.2 ± 3.3 mg GAE/g, respectively. In the case of the total flavonoid content (TFC), the results were slightly different, 90.22 ± 2.8 mg quercetin equivalent QE/g for UAE and 94.61 ± 1.9 mg QE/g for CE. However, there were no significant differences (*p* > 0.05) between the two methods.

### 2.2. Identification and Characterization of Phenolic Acid Derivatives, Flavonoids, and Lignanamides

Extract 1 and Extract 2 were further examined using Liquid Chromatography–Diode-Array Detection–Electrospray Ionization–Quadrupole Time-of-Flight Mass Spectrometry (LC-DAD-QToF-MS). Data were obtained using both positive and negative electrospray ionization source (ESI) modes, and it was observed that phenolic acids and their derivatives were ionized in the negative mode, whereas flavonoids were ionized in both positive and negatives modes. At the same time, lignanamides showed better ionization in the positive mode than in the negative mode. The identified compounds are summarized in Table 2, including the retention time, molecular formulas, *m/z* values in both positive and negative modes, and major high-resolution mass fragment ions. As displayed in Table 2, the extracts of the above-ground parts of *L. gmelinii* were composed of phenolic acid derivatives (sulfated) (1–10); flavonoids (11–36); lignanamides (37–46); and other compounds, such as amino acids, citric acids and their derivatives, hydroxyquinoline (quinoline derivative), and *N*-trans feruloyl tyramine (alkaloid) (47–54).

LC-DAD chromatograms are presented in Figure 1a at 254 nm, 280 nm, and 330 nm wavelengths, and representative LC-MS total current chromatograms in the positive and negative modes of ionization are presented in Figure 1b. No differences were observed in the chromatographic profiles between Extract 1 and Extract 2 of the above-ground parts of *L. gmelinii*.

Phenolic acids and sulfated derivatives (1–10): 

A total of 10 phenolic acids, such as gallic acid, protocatechuic acid, syringic acid, glucogallin, and their sulfated derivatives, were identified and tentatively characterized in the above-ground parts of *L. gmelinii*. The observed fragment ions are listed in Table 2.

Flavonoids (11–36):

Flavonoids were one of the major classes of compounds observed in the extracts. The tentative characterization of flavonoids was performed on the basis of our previous in-house report [18]. Gallocatechin, apigenin, quercetin, and myricetin derivatives, as well as their aglycone fragment ions, were the main flavonoids, as shown in Table 2.

Lignanamides (37–46):

Lignanamides are nitrogen-containing compounds, with characteristic fragment ion at m/z 287.0547 observed for limoniumin derivatives (38, 40–43) [3,19].

Other compounds (47–54):

In addition to the above-mentioned classes, the investigated extracts contained choline-*O*-sulfate, leucine and isoleucine amino acids, citric acids and their derivatives, a quinoline derivative, and N-trans feruloyl tyramine alkaloid.

### 2.3. Study of the Immunomodulatory Activity of the Extracts

#### 2.3.1. Effect of *L. gmelinii* Extracts on the Cytokine-Producing Activity of Murine Macrophages and Lymphocytes

Extracts 1 and 2 were dissolved in Dulbecco’s Modified Eagle Medium (DMEM) (Thermo Fischer Scientific, MA, USA) at the following concentrations: 1 mg/mL; 500; 100; 50; 25; 10; and 5 µg/mL. Splenic lymphocytes and peritoneal macrophages were obtained from male C57BL/6 mice to serve as target cells. Lymphocytes were activated with 5 µg/mL of Concanavalin A (ConA) (Sigma-Aldrich, St. Louis, MO, USA) and further cultured in an RPMI-1640 medium supplemented with Extracts 1 and 2 for 72 h. Peritoneal macrophages were stimulated with 500 pg/mL of lipopolysaccharides (LPSs) (Sigma-Aldrich, St. Louis, MO, USA), treated with the extracts, and incubated for 24 h at 37 °C and 5% CO_2_. The levels of cytokine production were measured using commercial enzyme-linked immunosorbent assay (ELISA) kits.

##### Effect of *L. gmelinii* Extracts on IL-6 Levels in Murine Macrophages and Lymphocytes

According to our results, presented in Figure 2a, Extracts 1 and 2 exerted a stimulatory effect on IL-6 production by non-activated macrophages at a concentration of 100 μg/mL to 5 μg/mL (Figure 2a). The maximum stimulatory effect on IL-6 production by non-activated macrophages was found at a concentration of 10 μg/mL of both extracts.

As expected, macrophages responded to LPS as a result of increased levels of IL-6 (747.73 pg/mL; 755.54 pg/mL), effectively suppressed by both extracts (Figure 2b). The maximum suppression of almost 3.5-fold in IL-6 production caused by macrophages was observed at the same concentration of 100 μg/mL of both extracts (Figure 2b).

Similar immunostimulatory and immunosuppressive effects of Extracts 1 and 2 were found on splenic lymphocytes. Treatment with the extracts caused an increase in IL-6 secretion in non-activated lymphocytes, as shown in Figure 3a. IL-6 production reached the maximum at a concentration of 5 μg/mL of Extract 1 and 10 μg/mL of Extract 2. On the contrary, the application of the extracts to ConA-activated splenic lymphocytes led to a significant decrease in the level of IL-6 (Figure 3b). The maximum inhibitory effect on IL-6 expression was caused by Extracts 1 and 2 at a concentration of 100 μg/mL (Figure 3b).

Thus, no significant difference in the effects of Extracts 1 and 2 on the level of IL-6 production was observed in both intact and activated macrophages and lymphocytes.

##### Effect of *L. gmelinii* Extracts on TNF-α Levels in Murine Macrophages and Lymphocytes

The effects of the *L. gmelinii* extract on TNF-α secretion in murine macrophages and lymphocytes were also studied. There was the same tendency towards an increase in the TNF-α level in intact cells and its reduction in LPS-stimulated macrophages and ConA-activated lymphocytes.

As shown in Figure 4a, Extracts 1 and 2 at the same concentration of 1 mg/mL significantly increased TNF-α secretion in intact macrophages to 2.41 ± 0.19 pg/mL and 2.65 ± 0.84 pg/mL, respectively. However, treatment of activated macrophages with Extracts 1 and 2 at a concentration of 1 mg/mL suppressed the level of TNF-α production (Figure 4b). No effects were observed at lower concentrations of the plant extracts.

When examining the level of TNF-α in non-activated lymphocytes, a statistically significant increase was observed at the same concentration of 1 mg/mL of both extracts. These results are shown in Figure 5a. However, when the splenic lymphocytes were stimulated with ConA, the extracts inhibited the secretion of TNF-α at all doses (Figure 5b). Treatment with Extracts 1 and 2 significantly decreased cytokine levels, with the maximum reduction at a concentration of 100 μg/mL.

#### 2.3.2. Effect of *L. gmelinii* Extracts on the Proliferation of Murine Splenic Lymphocytes

The proliferation of lymphocytes is a critical event in a proper immune response leading to the initiation and development of inflammation.

In this study, we determined the immunomodulatory effects of Extracts 1 and 2 on the proliferation of splenic lymphocytes in response to stimulation with ConA. An analysis of cell proliferation showed that Extract 2 more strongly suppressed the proliferation of activated splenic lymphocytes than Extract 1 (Figure 6a). A significantly high inhibition of the number of splenic lymphocytes (Extract 1, 65,750 ± 850) and (Extract 2, 32,835 ± 465) was produced at a dose of 1 mg/mL in comparison to the control (1,789,364 ± 511 and 1,788,125 ± 650, respectively).

However, both extracts had a stimulatory effect on the proliferation of non-activated lymphocytes (Figure 6b). The maximum number of proliferating cells was observed upon the addition of 50 µg/mL of Extract 1 and 25 µg/mL of Extract 2 (Figure 6b).

## 3. Discussion

The currently patented method for obtaining a valuable extract from the roots of *L. gmelinii* is 5 h maceration with 50% ethanol [24]. The extract obtained via this method contains a rich set of biologically active compounds: flavonols and their glycosides, condensed and hydrolysable tannins, pyrogallol, gallic and ellagic acids, proanthocyanidins, polyphenolic compounds, and previously undescribed forms of flavan-3-ols [15,16,17,18,25]. As such, polyphenols of botanic origin have been proven to act as immunomodulators [26]. The aim of the current research was to study the immunomodulatory activity of extracts from the above-ground parts of *L. gmelinii*, as well as to examine how ultrasonic-assisted treatment affects the extraction process and chemical composition of the obtained extracts.

It is reported that UAE can serve as a more efficient and environmentally friendly method due to its short processing time, lower consumption of solvent and energy, less environmental pollution, and low maintenance cost of equipment [27]. Moreover, several studies have shown an increase in the yield of extracts and their activity when using ultrasound in comparison with other methods [28,29]. At the same time, some authors have noted that the yields of the secondary metabolites obtained when using UAE are lower than or equal to those obtained when using CE [30].

In our study, the efficiency of the CE and UAE methods of extraction was assessed via the yield of extracts and the content of the main secondary metabolites in them (Table 1), and it was found that Extracts 1 and 2 differed insignificantly. The TPC values were 378.1 ± 4.5 mg GAE/g DW in Extract 1 and 382.2 ± 3.3 mg GAE/g DW in Extract 2, and these are much higher than those reported for *L. gmelinii* extracts from other countries, at 54.42 mg GAE/g DW [31] and 5.60 ± 0.45 mg GAE/g DW [32]. In reports [31] and [32], the TFC values as mg of catechin equivalent (CE) were a little over 2 mg CE/g DW and 1.57 ± 0.10 mg CE/g DW, respectively, which is significantly lower than those in our results. This may be due to the impact of geographical origin, environmental and seasonal variations [33], and the methods and conditions of extraction on phytochemical composition, which may alter the amount of phenolics [27].

As stated above, the polyphenolic complex is supposed to be a predominant class in the hydroethanolic extracts of *L. gmelinii*, with the biological activity possibly attributed to the presence of these secondary metabolites. The chemical constituents of *L. gmelinii* detected using LC-QToF-MS were phenolic acid derivatives (sulfated) (1–11); flavonoids (12–43); lignanamides (44–53); and other compounds, such as amino acids, citric acids and their derivatives, hydroxyquinoline (quinoline derivative), and N-trans feruloyl tyramine (alkaloid) (54–62). No differences in the chromatographic profiles were observed among the CE and UAE extracts from the above-ground parts of *L. gmelinii.* The obtained extracts displayed complex chemical compositions, which led to the fair assumption that they may have immunomodulatory activity.

It was also of interest to determine whether there was a difference in the biological activities of the extracts. The in vitro effects of the *L. gmelinii* extracts on the induction of murine macrophages and splenic lymphocytes that produce proinflammatory cytokines (IL-6 and TNF-α) and their action on the proliferative responses were studied for the first time.

There have been numerous studies on the immunomodulatory activity of phenolic-rich plant extracts and dietary polyphenols, and their data are in good agreement with the data obtained in this study. An immunosuppressive action linked to the downregulation of the gene expression of proinflammatory cytokines associated with the presence of polyphenolic and flavonoid constituents was suggested for a number of plants [34,35,36]. On the contrary, some reports showed an immunostimulatory effect of phenolic-rich extracts on the production of proinflammatory cytokines. As such, the MeOH extract of *P. campechiana* boosted the production of IL-6 and TNF-α by murine macrophages in a concentration-dependent manner [37]. Another study demonstrated that a herbal composition rich in phenols and flavonoids increased TNF-α and decreased IL-10 and IL-6 levels in CP-induced immunosuppressed mice [38].

Extracts 1 and 2 isolated from *L. gmelinii* effectively suppressed the increased levels of TNF-α and IL-6 in LPS- and ConA-stimulated macrophages and lymphocytes, respectively. At the same time, the treatment of non-activated macrophages and lymphocytes with Extracts 1 and 2 significantly increased the secretion of cytokines. The same stimulatory or inhibitory effects of Extract 1 and Extract 2 on lymphocyte proliferation depending on their state were also verified. However, no significant difference was noted in the immunomodulatory activity of the *L. gmelinii* extracts, supported by the above data on chemical composition. In similar studies of the immunomodulatory effects of *Limonium* species, it was demonstrated that the polysaccharides of *L. sinense* improved the phagocytosis activity of macrophages in immunosuppressed mice and increased the proliferation and production of IFN-γ and IL-2 by splenocytes in vitro [11]. The extract of *L. duriusculum* and the isolated flavonoid apigenin inhibited proinflammatory NF-κB-dependent transcriptional responses in HCT116 and THP-1 cells [39]. These observations, as well as the results of the current study, raise the prospect of using the extracts of *L. gmelinii* and related species for the treatment of diseases associated with immune disorders.

## 4. Conclusions

The results of the current study demonstrate that the method of obtaining extracts (CE and UAE) from the above-ground parts of *L. gmelinii* does not significantly affect the yield, TPC, TFC, chemical composition, or immunomodulatory activity. Therefore, UAE could have advantages over CE, as it is significantly faster and simpler and has a lower solvent consumption and “greener” approach [27]. The LC-QToF-MS analysis clearly showed impressive compositions of both extracts, rich in biologically active compounds with remarkable immunomodulatory potential, such as apigenin [39], epigallocatechin-3-gallate [20] and quercetin [20]. This work reported, for the first time, that extracts from the above-ground parts of *L. gmelinii* may modulate the inflammatory mode of murine macrophages and lymphocytes via the stimulation or suppression of TNF-α and IL-6 production and splenocyte proliferation depending on the status of these immune cells. The data obtained provide possibilities for further research into the mechanisms of the immunomodulatory activity of *L. gmelinii* extracts, which could be recommended for further preclinical and clinical studies.

## 5. Materials and Methods

### 5.1. Chemicals and Reagents

Ethanol ≥96% was purchased from Talgarspirt. Folin–Ciocalteu reagent, gallic acid ≥96%, and quercerin ≥95% were purchased from Sigma-Aldrich. Sodium carbonate and aluminum chloride were bought from AppliChem GmbH. LPS (from *Escherichia coli* 0111:B4) was from Sigma-Aldrich, and DMEM high-glucose medium was from Gibco. The used acetonitrile, methanol, and formic acid were of HPLC-certified grade, and water was purified using a Milli-Q system (Millipore).

### 5.2. Plant Raw Materials

The above-ground parts of *L. gmelinii* were collected in the territory of the Almaty region during the flowering period of 2020 and identified at the Institute of Botany and Phytointroduction in Almaty, Kazakhstan, with the voucher specimen number 154,958. During the trip to the territory of the Zhanalyk village of the Talgar region, the coordinates of the place of the collection and the growth of *L. gmelinii* species were noted (Table 3), and raw material was collected for the study and production of extracts. The raw materials were identified for authenticity using macro- and microscopy methods upon independent assessment at the laboratories of the Institute of Botany and Phytointroduction and al-Farabi Kazakh National University (Almaty, Kazakhstan), upon which they were thoroughly cleaned and dried and subsequently milled to a particle size of 3 mm. Quality control of plant raw materials was carried out in accordance with the requirements of the State Pharmacopeia of the Republic of Kazakhstan.

### 5.3. Preparation of Extracts and Their Standardization

#### 5.3.1. Maceration and Ultrasonic-Assisted Extraction

To obtain the extract via UAE, the extraction of the milled plant material was carried out once under the optimum conditions [40] in an Elmasonic S 450 ultrasonic bath (Elma Electronic UK Ltd., Bedford, UK) at a ratio of raw material to solvent of 1:5 for 45 min at 30 °C. The filtered aqueous–alcoholic extract was concentrated using a rotary evaporator, model IKA RV20 (IKA^®^-Werke GmbH & Co. KG, Staufen im Breisgau, Germany), at a temperature of 40–45 °C in a vacuum to dryness until a brown crystalline powder was obtained.

To obtain the extract via CE, 5 h double extraction was carried out with a six-fold excess of 50% ethanol at a temperature of 20–23 °C, and the plant material was subsequently washed with water to remove the extract residues. The first and second extracts were mixed and concentrated to dryness under mild conditions on a rotary evaporator to obtain a dry extract. The extract yield was calculated using the formula:$$Yield,\%=\frac{Weight\;of\;dried\;extract}{Weight\;of\;plant\;material}\times 100$$

#### 5.3.2. Total Phenolic Content

The total phenolic contents (TPC) in the extracts were determined using the Folin–Ciocalteu reagent in accordance with the methodology in [41] with some modifications. About 200.0 mg of the substance was transferred into a 100.0 mL volumetric flask and quantitatively dissolved in 30% ethanol. Then, a 2.0 mL aliquot was taken and diluted with distilled water in a 100.0 mL volumetric flask.

Next, 1.0 mL of this solution was mixed with 1.0 mL of the Folin–Ciocalteu reagent and 15.0 mL of distilled water and left for 6 min. Subsequently, 3.0 mL of 20.0% sodium carbonate solution was added, and the flask was placed in a dark place. After 120 min, the absorbance corresponding to the total amount of polyphenols was measured at 760 nm on a JASCO J-715 spectrophotometer (Jasco Corp., Tokyo, Japan) using distilled water as a reference solution. Gallic acid was used as a standard for plotting a calibration line.

#### 5.3.3. Total Flavonoid Content

The assessment of the total flavonoid content was carried out according to the method described in [42] with some modifications; quercetin was used as a standard. The dry extract was dissolved in 100.0 mL of 90% ethanol to prepare a solution with a concentration of 1.0 mg/mL. Then, 2.0 mL of a sample solution and 1.0 mL of 1% aluminum chloride in 95% ethanol were mixed and diluted to 25.0 mL in a volumetric flask. After 20 min, the optical density of the solution was measured on a spectrophotometer (Jasco Corp., Tokyo, Japan) at a wavelength of 415 nm. A solution consisting of 2.0 mL of the sample solution was diluted to 25.0 mL in a volumetric flask with 95% ethanol and used as a reference.

### 5.4. Chemical Characterization of Extract from Above-Ground Part of L. gmelinii Using Liquid Chromatography–Diode-Array Detector–Quadrupole Time-of-Flight Mass Spectrometry (LC-DAD-QToF)

#### 5.4.1. Sample Preparation

About 20 mg of the extract from the above-ground parts of *L. gmelinii* was sonicated in 1.0 mL of methanol for 5 min followed by centrifugation for 10 min at 5000× *g* rpm separately. The supernatant-filtered solution was used for analyses.

#### 5.4.2. Instrumentation and Analytical Conditions

The LC-QToF-MS method was used for the analysis of the extracts from the above-ground samples of *L. gmelinii* using the reported method [18].

### 5.5. In Vivo Experiments to Determine Immunomodulatory Activity

#### 5.5.1. Preparation of the Test Specimens

Extracts 1 and 2 were dissolved in culture medium DMEM at the following 7 concentrations: 1 mg/mL; 500 µg/mL; 100 μg/mL; 50 μg/mL; 25 µg/mL; 10 μg/mL; and 5 µg/mL. The diluted solutions were sterilized using a polyethersulfone membrane filter with a pore size of 0.2 μm (TPP).

#### 5.5.2. Animals

Male C57BL/6 10–12-week-old mice were purchased from Masgut Aikimbayev’s National Scientific Center for Especially Dangerous Infections (Almaty, Kazakhstan). The animals were housed in a temperature-controlled environment (23 °C) with 60% relative humidity applying a 12 h light/dark cycle. The animals had ad libitum access to food and water. The experimental procedures involving animals were in full compliance with the current international laws and policies (Guide for the Care and Use of Laboratory Animals, National Academy Press, 1996) [43] and were approved by the Local Ethics Committee for Animal Use in the National Center for Biotechnology (Protocol No. 4 from 8 September 2020).

#### 5.5.3. Isolation of Lymphocytes

The animal was sacrificed via cervical dislocation and treated with a 70% ethanol solution; the abdomen was carefully cut, and the spleen was removed. The spleen was immediately transferred to a Petri dish with DMEM for washing. After this procedure, the spleen was transferred to a glass homogenizer filled with DMEM (4–5 mL) and homogenized with a Teflon pestle. The homogenate containing mainly lymphocytes was carefully removed with a Pasteur pipette and transferred into a sterile centrifuge tube (15 mL). Splenocytes were pelleted via centrifugation at 1500 rpm for 5 min. To remove erythrocytes, 3 mL of BD Pharm Lyse^®^ lysing solution (BD Biosciences, Dubai, UAE) was added to the cell pellet and kept for 5 min on ice. After the hemolysis of erythrocytes, the splenocytes were washed three times with large volumes of DMEM to completely remove the lysing solution. Then, 2 mL of culture medium RPMI-1640 containing 10% fetal bovine serum (FBS) was added to the cell pellet and thoroughly resuspended with a vortex (Vortex 1, IKA-Werke GmbH & Co. KG, Staufen im Breisgau, Germany). The cells were counted with a TC20^®^ automatic cell counter (Bio-Rad, Hercules, CA, USA).

#### 5.5.4. Isolation of Peritoneal Macrophages

The animal was sacrificed via cervical dislocation and treated with a 70% ethanol solution. The skin was removed from the abdomen without piercing the abdominal wall. Then, 3 mL of DMEM was injected with some air into the abdominal cavity. In this case, the abdomen of the mouse swells, tightening the injection site. Then, using surgical forceps or a spatula, the abdominal wall was massaged for 1 min to separate peritoneal macrophages into suspension. Using a syringe, fluid was collected from the lateral side of the abdominal cavity and then transferred into sterile centrifuge tubes.

Peritoneal macrophages were isolated from other cells via their high adhesivity to a plastic surface. For this, peritoneal cells were suspended at a concentration of 2 × 10^6^ cells/mL in RPMI-1640 medium containing 10 mM HEPES and 10% FBS, placed in polystyrene 24-well plates at 1 mL per well, and left for 1 h at 37 °C and 5% CO_2_. Then, the non-adherent cells were carefully removed, and the monolayer of macrophages was washed once with the DMEM. Cell viability was determined using 0.1% trypan blue. Macrophages were incubated in 1 mL of RPMI-1640 culture medium for 24 h at 37 °C and 5% CO_2_.

#### 5.5.5. Determination of Cytokine Levels

The measurement of cytokine levels was performed using ELISA. ELISA was performed using the following comsigmial kits: a Mouse TNF alpha SimpleStep ELISA kit and an IL-6 Mouse SimpleStep ELISA kit (all ELISA kits were from Abcam, Cambridge, UK). All procedures were performed according to the manufacturer’s instructions.

#### 5.5.6. Lymphocyte Proliferation Assay

Extracts 1 and 2 were dissolved in a complete nutrient medium, DMEM or RPMI-1640, at the following 7 concentrations: 1 mg/mL; 500 µg/mL; 100 μg/mL; 50 μg/mL; 25 µg/mL; 10 μg/mL; and 5 µg/mL. The splenic lymphocytes of C57BL/6 mice served as target cells. Prior to the addition of the preparations, lymphocytes were activated with ConA (Sigma-Aldrich) at a concentration of 5 µg/mL, after which they were cultivated in RPMI-1640 medium with the addition of Extracts 1 and 2 for 72 h. Splenic lymphocytes were stained with anti-Ki67 antibodies and analyzed using a Muse cellular analyzer (MilliporeSigma, Darmstadt, Germany).

### 5.6. Statistical Processing of Data

All data are presented as mean ± standard deviation. Statistical significance was calculated using a one-way analysis of variance (ANOVA) test. *p* < 0.05 was considered statistically significant. The statistical analysis was conducted with software Statistica 6.0 (StatSoft, Tulsa, OK, USA).

## Figures and Tables

**Figure 1 plants-12-04019-f001:**
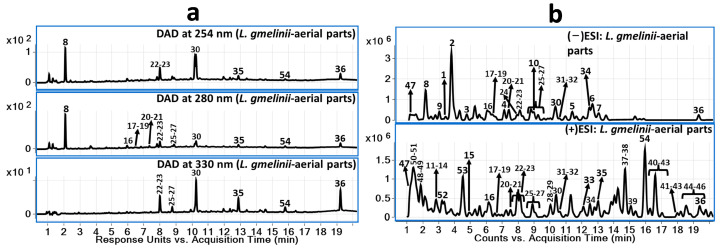
(**a**) LC-DAD chromatograms at 254, 280, and 330 nm of the above-ground parts of *L. gmelinii*; (**b**) LC-QToF-MS chromatograms in negative and positive modes along with DAD chromatograms.

**Figure 2 plants-12-04019-f002:**
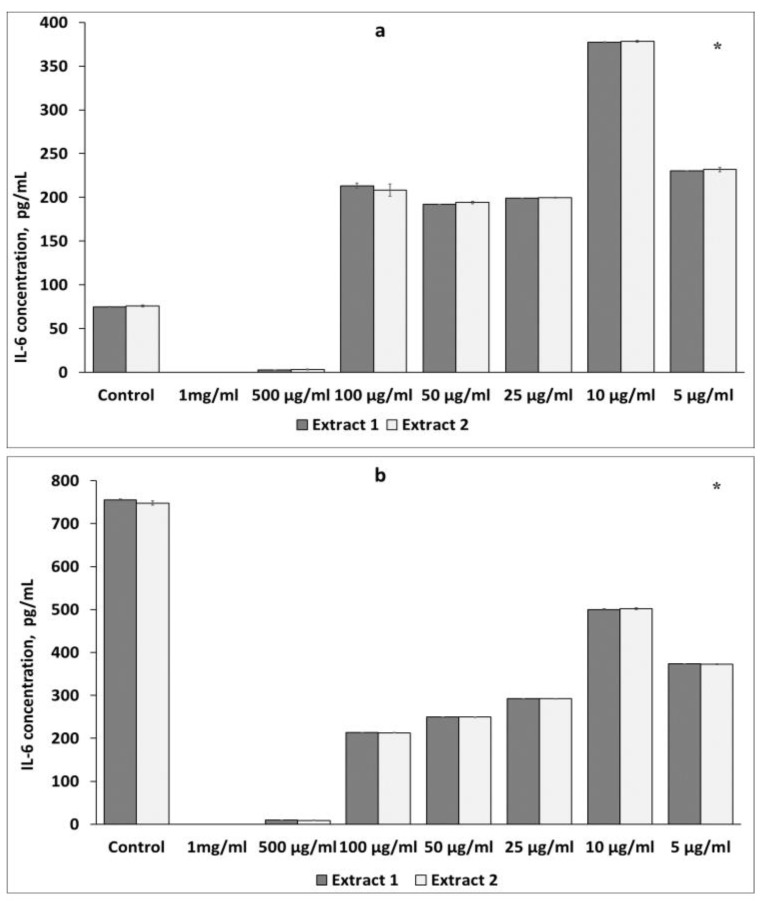
Effects of *L. gmelinii* extracts on the secretion of IL-6 by non-activated (**a**) and activated (**b**) macrophages. Controls were used in the experiment: macrophages without treatment or stimulus and macrophages activated with LPS. Data represent mean ± S.D (*n* = 6). (*) indicates significant differences in comparison to control; *p* < 0.05 in comparison to control.

**Figure 3 plants-12-04019-f003:**
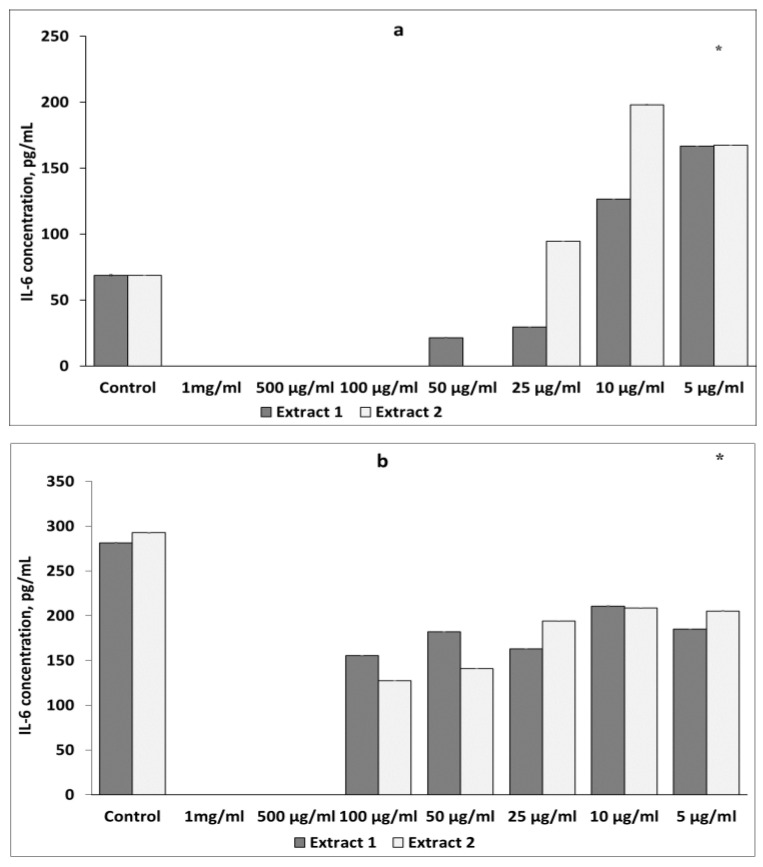
Effects of *L. gmelinii* extracts on the secretion of IL-6 by splenic T lymphocytes without treatment or stimulus (**a**) and by splenic T lymphocytes activated with ConA (**b**). Controls were used in the experiment: splenic T lymphocytes without treatment or stimulus and splenic T lymphocytes activated with ConA. Data represent mean ± S.D (*n* = 6). (*) indicates significant differences in comparison to control; *p* < 0.05 in comparison to control.

**Figure 4 plants-12-04019-f004:**
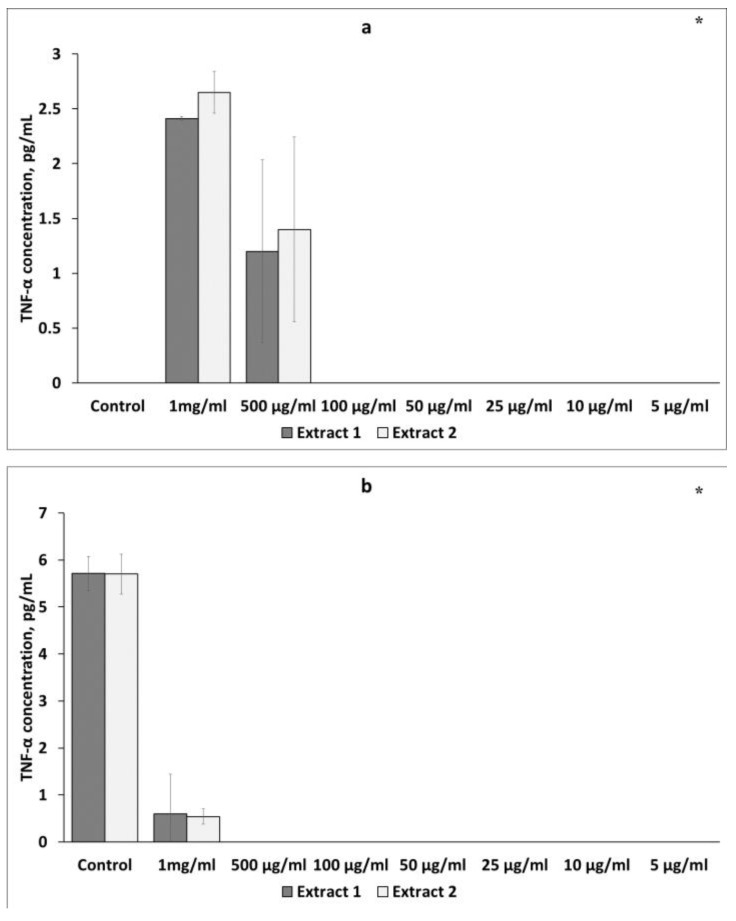
Effects of *L. gmelinii* extracts on TNF-α secretion by intact (**a**) and activated (**b**) macrophages. Controls were used in the experiment: macrophages without treatment or stimulus and macrophages activated with LPS. Data represent mean ± S.D (*n* = 6). (*) indicates significant differences in comparison to control; *p* < 0.05 in comparison to control.

**Figure 5 plants-12-04019-f005:**
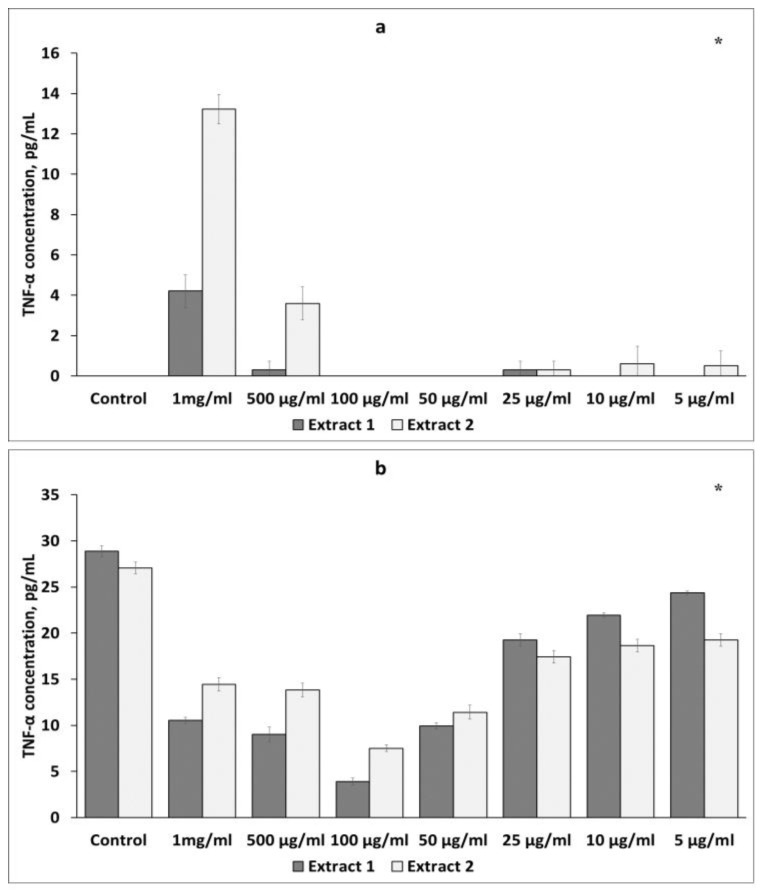
Effects of *L. gmelinii* extracts on the secretion of TNF-α by splenic T lymphocytes without treatment or stimulus (**a**) and splenic T lymphocytes activated with ConA (**b**). Controls were used in the experiment: splenic T lymphocytes without treatment or stimulus and splenic T lymphocytes activated with ConA. Data represent mean ± S.D (*n* = 6). (*) indicates significant differences in comparison to control; *p* < 0.05 in comparison to control.

**Figure 6 plants-12-04019-f006:**
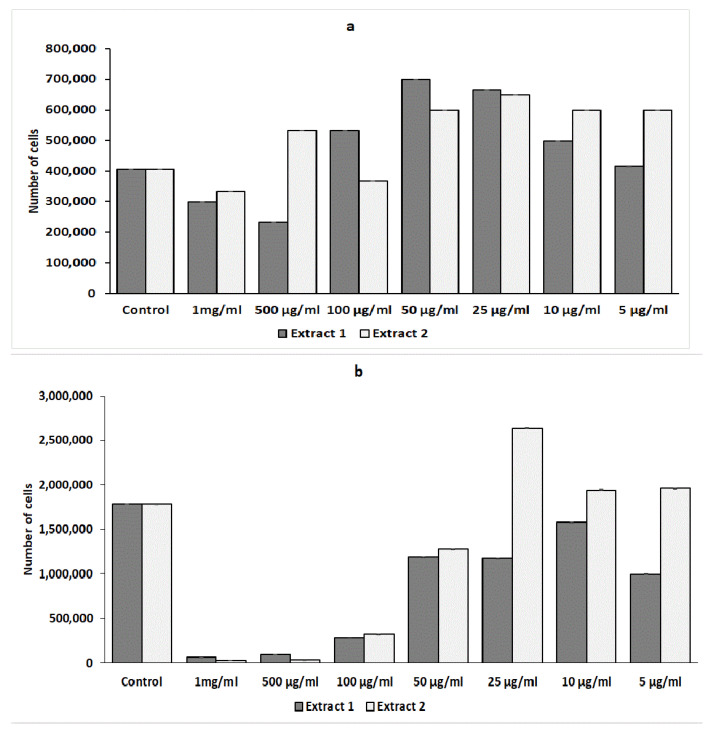
Effect of *L. gmelinii* extracts on the proliferation of splenic T lymphocytes without treatment or stimulus (**a**) and spleen splenic T lymphocytes activated with Con A (**b**). Controls used in the experiment: splenic T lymphocytes without treatment or stimulus and T lymphocytes activated with ConA. Data represent mean ± S.D (*n* = 6).

**Table 1 plants-12-04019-t001:** Yield and content of secondary metabolites in *L. gmelinii* extracts (mean  ±  SD, n  =  3).

	Extract 1 (UAE)	Extract 2 (CE)
Yield ^a^, %	30.2 ± 1.6	33.5 ± 2.4
The total amount of polyphenols ^a^, mg GAE/g	378.1 ± 4.5	382.2 ± 3.3
The total amount of flavonoids ^a^, mg QE/g	90.22 ± 2.8	94.61 ± 1.9

^a^ Results are presented as mean ± standard deviation.

**Table 2 plants-12-04019-t002:** Table 2. Tentative identification and characterization of natural compounds in extracts from aboveground parts of *L. gmelinii* using LC-QToF in positive and negative ionization modes.

#	RT (Min)	Tentative Compound Name	Molecular Formula	Mass	[M+H]^+^	Fragment ions (Positive ion Mode)	[M-H]^−^	Fragment ions (Negative ion Mode)	Above-the-Ground Parts	
									Macer-ation	UAE
Sulfated phenolic acids										
1	3.40	Sulfosalidroside	C_14_H_20_O_10_S	380.0777	-	-	379.0704 (379.0704) *	96.9602	+	+
2	3.71	Sulphonic acid	C_8_H_10_O_5_S	218.0249	-	-	217.0184 (217.0176)	96.9605	+	+
3	4.66	*p*-Coumaroyl-(β-D-glucose-6-*O*-sulfate)	C_15_H_18_O_11_S	406.0570	-	-	405.0506 (405.0497)	247.0291, 241.0034, 163.0404, 96.9607	+	+
4	7.09	dihydromyricetin-3′-*O*-sulfate	C_15_H_12_O_11_S	400.0100	-	-	399.0032 (399.0028)	319.0450, 151.0035, 137.0245	+	+
5	11.41	Deoxy-dihydromyricetin-3′-*O*-sulfate	C_15_H_12_O_10_S	384.0151	-	-	383.0083 (383.0078)	151.0042	+	+
6	12.70	Myricetin-3′-*O*- sulfate	C_15_H_10_O_11_S	397.9944	-	-	396.9874 (396.9871)	317.0308, 151.0039, 137.0246	+	+
7	13.09	Pinoresinol-4-sulfate	C_20_H_22_O_9_S	438.0985	-	-	437.0920 (437.0912)	396.9873, 380.9929	+	+
Phenolic acid/derivatives										
8	2.04	Gallic acid	C_7_H_6_O_5_	170.0215	-	-	169.0148 (169.0142)	125.0250	+	+
9	2.90	Protocatechuic acid	C_7_H_6_O_4_	154.0266	-	-	153.0202 (153.0193)	109.0299	+	+
10	8.94	Syringic acid	C_9_H_10_O_5_	198.0528	-	-	197.0459 (197.0455)	169.0148	+	+
Flavonoids										
11	2.84	Gallocatechin/Epigallo-catechin/3,5,7,3′,4′,6′-hexahydroxyflavan	C_15_H_14_O_7_	306.074	307.0812 (307.0812)	139.0382	305.0669 (305.0667)	137.0246	+	+
12	3.24								+	+
13	4.00								+	+
14	4.93								+	+
15	5.00	(−)-Epigallocatechin-(4β→8)-(-)-epigallocatechin-3-*O*-gallate	C_37_H_30_O_18_	762.1432	763.1508 (763.1505)	307.0817, 139.0392	761.1358 (761.1359)	423.0715	+	+
16	6.07	Epigallocatechin-(2β→7,4β→8)-epigallocatechin-3-*O*-gallate	C_37_H_28_O_18_	760.1276	761.1348 (761.1348)	287.0544, 139.0389	759.1200 (759.1203)	453.1038	+	+
17	6.59	3-*O*-Galloylepigallo-catechin (4β→8)-epigallocatechin-3-*O*-gallate/Theasinensin A/Theasinensin D	C_44_H_34_O_22_	914.1542	915.1607 (915.1614)	745.1371, 595.1069, 457.0760, 287.0555, 153.0186	913.1477 (913.1469)	423.0729, 285.0405	+	+
18	6.98								+	+
19	9.75								+	+
20	6.72	Epigallocatechin gallate/Gallo-epicatechin gallate	C_22_H_18_O_11_	458.0849	459.0926 (459.0922)	289.0711, 139.0393	457.0777 (457.0776)	305.0681, 169.0157	+	+
21	7.45								+	+
22	7.83	Myricetin 7-(6′/6″-galloylglucoside)	C_28_H_24_O_17_	632.1013	633.1089 (633.1086)	481.0966, 319.0446, 287.0545, 153.0180	631.0947 (631.0941)	559.0402, 479.0836, 317.0307	+	+
23	8.13					319.0455, 153.0182		479.0832, 316.0226	+	+
24	7.91	Samarangenin B	C_44_H_32_O_22_	912.1385	913.1449 (913.1458)	743.1225, 617.0932, 455.0596	911.1305 (911.1312)	-	+	+
25	8.67	Myricetin 3-galactoside (Gmelinoside I)/ Myricetin 3-β-D-sorboside/Myricetin 3-glucoside	C_21_H_20_O_13_	480.0904	481.0974 (481.0977)	319.0451	479.0836 (479.0831)	316.0235	+	+
26	8.91								+	+
27	9.10								+	+
28	9.46	Myricetin 3-xyloside	C_20_H_18_O_12_	450.0798	451.0871 (451.0871)	319.0453	449.0727 (449.0725)	316.0229	+	+
29	10.05								+	+
30	10.38	3,3′,4′,5,5′,7-Hexahydroxyflavone; 3-*O*-[rhamnopyranosyl-(1→2)-rhamnopyranoside]	C_27_H_30_O_16_	610.1534	611.1607 (611.1606)	465.1043, 319.0461	609.1461 (609.1461)	463.0893, 316.0233	+	+
31	10.60	Quercetin glucoside	C_21_H_20_O_12_	464.0955	465.1030 (465.1028)	303.0497	463.0881 (463.0882)	301.0346	+	+
32	10.86								+	+
33	12.2	Myricetin	C_15_H_10_O_8_	318.0376	319.0453 (319.0448)	153.0183	317.0310 (317.0303)	151.0041, 137.0246	+	+
34	12.50	Quercetin rhamnoside (Quercitrin)	C_21_H_20_O_11_	448.1006	449.1080 (449.1082)	303.0504	447.0929 (447.0933)	301.0354	+	+
35	12.99	Apigenin-7-glucuronide	C_21_H_18_O_11_	446.0849	447.0925 (447.0922)	271.0611	445.0780 (445.0776)	269.0456	+	+
36	19.37	Apigenin	C_15_H_10_O_5_	270.0528	271.0603 (271.0601)	153.0185, 119.0489	269.0460 (269.0455)	117.0351	+	+
Lignanamides										
37	14.92	Cannabisin A	C_34_H_30_N_2_O_8_	594.2002	595.2081 (595.2075)	-	-	-	+	+
38	14.95	Limoniumin A	C_26_H_19_NO_7_	457.1162	458.1236 (458.1234)	287.0550	-	-	+	+
39	15.14	Cannabisin B/3,3′-Demethyl-heliotropamide	C_34_H_32_N_2_O_8_	596.2159	597.2235 (597.2231)	295.0599, 187.0391, 153.0181	-	-	+	+
40	16.40	Limoniumin B	C_27_H_21_NO_7_	471.1318	472.1382 (472.1391)	287.0547	-	-	+	+
41	16.80	Limoniumin F/H/I/Cannabisin C	C_35_H_34_N_2_O_8_	610.2315	611.2383 (611.2388)	287.0546	-	-	+	+
42	17.10								+	+
43	17.73								+	+
44	18.52	Cannabisin D/F	C_36_H_36_N_2_O_8_	624.2472	625.2544 (625.2544)	367.2091	-	-	+	+
45	18.81								+	+
46	19.20								+	+
Others										
47	1.11	Choline-*O*-sulfate	C_5_H_13_NO_4_S	183.0565	184.0638 (184.0638)	104.1072 (Choline)	182.0487 (182.0493)	-	+	+
48	1.20	Leucine/Isoleucine (Amino acids)	C_6_H_13_NO_2_	131.0946	132.1022 (132.1019)	86.0957	-	-	+	+
49	1.81								+	+
50	1.32	Citric acid/Isocitric acid	C_6_H_8_O_7_	192.0270	-	-	191.0197 (191.0202)	111.0089	+	+
51	1.41								+	+
52	3.10	2-Ethyl citrate	C_8_H_12_O_7_	220.0583	-	-	219.0520 (219.0513)	183.0305, 111.0086	+	+
53	4.51	Hydroxyquinoline (Quinoline derivative)	C_9_H_7_NO	145.0528	146.0604 (146.0600)	128.0495, 91.0546, 77.0391	-	-	+	+
54	15.95	*N*-*trans*-feruloyl tyramine (Alkaloid)	C_18_H_19_NO_4_	313.1314	314.1385 (314.1381)	177.0541, 145.0281, 117.0330	-	-	+	+

* theoretical exact mass; ‘+’ indicates presence of the compound and ‘ND’ indicates the compound not detected in respective plant part extract; ‘UAE’ indicates Ultrasound-assisted extraction.

**Table 3 plants-12-04019-t003:** Coordinates of the collection of plant materials.

Species	Collection Area	GPS Coordinates	Phenophase	Collected Part	Phytomass, kg
*Limonium gmelinii,* aboveground parts	Talgar region, Zhanalyk village	H = 1326 m above the sea level N = 43°34′28″ E = 077°03′20″	budding, beginning of flowering. sp-sol, blooming	Above ground part (fresh)	10.5

## Data Availability

Data are contained within the article.

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
