# Peer review of "Phytochemical Profiles and In Vitro Immunomodulatory Activities of Extracts Obtained from Limonium gmelinii Using Different Extraction Methods"

_plants, 2023, doi:10.3390/plants12234019_

Round 1

Reviewer 1 Report

The authors should modify the title to Comparison of yield, chemical composition and immunomodulatory activity of extracts obtained from Limonium gmelinii (Willd.) Kuntze by different extraction methods.

Introduction should be modified and shortened to necessary information closely related to the research.

All figures should be modified by describing them precisely and statistical designations should be corrected. 

Many sections in materials and methods should be clarified.

Overall, the manuscript appears chaotic and disorganised with particular emphasis on the very imprecise preparation of the figures.

The manuscript should be linguistically corrected.

Reviewer 2 Report

The subject of the present paper is interesting, the work is well designed, the obtained results are relevant, and seems to have novelty. The manuscript is worth to be published provided that the issues pointed out below can be conveniently addressed.

What the authors want to mean with Limonium (L.) gmelinii?

The manuscript requires editing of English language and style.

There must be consistency on the Abbreviations should be defined the first time they are used (e.g. UAE, CE, TPC, GAE, QE; etc.)

Table 1: Superscripts a, b and c mean the same. So, only one is sufficient.

Data on table 2 is relevant enough to be insert in the body of the paper. Moreover, it will help the reading of section 2.2.

It is not clear what parts of the plant were used. In Line 139 it is referred “the aerial parts and root extracts”; in table 2 only the aerial parts are mentioned; in Section 4.2. (Plant raw materials) no information is provided concerning the parts used; in section 4.4.2 it is referred that LC-QToF-MS was used for the analysis of extracts from aboveground Limonium.

 In Figures 2, 3, 4, 5, 6 and 7 caption’s it is unnecessary to describe the origin of Extracts 1 and 2.

Paragraph from line 272 to 285 is too general and is not suitable for the Discussion od Results; maybe it can be merged in the introduction.

Paragraph from line286 to 301 seems more a review paragraph than a discussion one. It should be removed or shortened.

Discussion of the immunomodulatory activity of the extracts, which is an important part of the work, is very superficial and it should be rewritten.

Conclusions do not refer conveniently the immunomodulatory activity of the extracts.

Reference 35 is incomplete.

The manuscript requires editing of English language and style.

Reviewer 3 Report

The manuscript "Comparison of yields, chemical composition and immunomodulatory activity of extracts isolated from Limonium gmelinii (Willd.) Kuntze by different extraction methods" was well written. The results are congruent with de conclusions. However, some points should be improved. 

1. Revise the language;

2. Write in full the first time you use an acronym;

3. In the figure captions, indicate the number of experiments and replications.

Minor editing of English language required

Round 2

Reviewer 1 Report

I accept in its present form.

.